# Pancreatic Injury after COVID-19 Vaccine—A Case Report

**DOI:** 10.3390/vaccines9060576

**Published:** 2021-06-01

**Authors:** Artur Cieślewicz, Magdalena Dudek, Iwona Krela-Kaźmierczak, Anna Jabłecka, Maciej Lesiak, Katarzyna Korzeniowska

**Affiliations:** 1Department of Clinical Pharmacology, Poznan University of Medical Sciences, 61-861 Poznan, Poland; ajablecka@ump.edu.pl (A.J.); katakorz@wp.pl (K.K.); 21st Department of Cardiology, Poznan University of Medical Sciences, 61-848 Poznan, Poland; magdalena.dudek@skpp.edu.pl (M.D.); maciej.lesiak@ump.edu.pl (M.L.); 3Department of Gastroenterology, Dietetics and Internal Diseases, Poznan University of Medical Sciences, 60-355 Poznan, Poland; ikrela-kazmierczak@ump.edu.pl

**Keywords:** pancreatitis, pancreatic injury, COVID-19, coronavirus vaccine, Comirnaty, COVID-19 mRNA vaccine

## Abstract

The COVID-19 pandemic has caused more than 3 million deaths worldwide. Recently developed genetically engineered vaccines are the most critical solution for controlling the pandemic. Clinical trials on a large number of participants confirmed their safety and efficacy. However, with the growing number of vaccinated people, new infrequent adverse effects have been reported, not described in the medicinal product characteristics. We would like to report a case of acute pancreatic injury that occurred shortly after administering Pfizer BioNTech COVID-19 mRNA vaccine (Comirnaty). The report points out the potential need for close monitoring of patients reporting abdominal pain after vaccination (unresponsive to standard oral painkillers) because such symptom can be associated with acute pancreatitis.

## 1. Introduction

The COVID-19 pandemic has already caused more than 3 million deaths worldwide [1]. An essential method for controlling the pandemic has recently been provided by vaccines—modern genetically engineered preparations designed to trigger the immune response to coronavirus antigen. A few such vaccines have been registered by the FDA and EMA [2,3,4,5]. Their safety and efficacy were assessed on large groups of participants (the combined number of recruits was >130,000). The most common adverse effects were similar in all preparations and included pain at the injection site, fatigue, headache, myalgia, arthralgia, pyrexia and nausea [2,3,4,5]. With the growing number of vaccinated people, more and more adverse effects are being reported, including adverse events not described in medicinal product characteristics. The report we would like to present describes a very rare case of pancreatic injury and suspected acute pancreatitis that occurred shortly after administering the Pfizer BioNTech COVID-19 mRNA vaccine (Comirnaty) to a young and healthy patient.

## 2. Case

The patient, an MD and a healthcare worker, is a 29-year-old female Caucasian with no history of pancreatitis, concomitant diseases and allergic reactions to drugs or vaccines. She is a healthy woman, breastfeeding mother who did not consume alcohol or drugs. On 8 January 2021, she was administered the first dose of Pfizer BioNTech COVID-19 mRNA vaccine. A few minutes after receiving the vaccine, she reported severe pain at the injection site, radiating to her left hand and neck.

Twelve hours after vaccination, the patient experienced muscle pain, headache, chills and general weakness, which lasted for about 3 h. Twenty hours after vaccination, she woke up in the night because of severe abdominal pain. Despite taking paracetamol (2 × 500 mg, orally), the pain level increased over the next hours and became radiating to the spine. Twenty-eight hours after vaccination, the patient still suffered from severe pain in the upper abdomen, unresponsive to standard oral painkillers. Moreover, fever up to 40 °C occurred. The timeline of adverse effects is presented in Figure 1.

The patient went to the hospital, suspecting pancreatitis. Laboratory analyses were carried out the next day after vaccination. Blood morphology was normal (Table 1), except for minor disturbances in neutrophil and lymphocyte percentage, slightly increased mean corpuscular haemoglobin concentration (MCHC) and increased platelet anisocytosis (PDW). Biochemical analysis revealed significantly increased CRP and urine amylase (Table 2). Nasopharyngeal swab for RT-PCR COVID-19 testing taken in Emergency Room was negative. Abdominal ultrasonography was performed, resulting in hyperechoic lesion in the right lobe of the liver, gallbladder not enlarged with normal wall, intrahepatic bile ducts not dilated, pancreas clearly visible in the head and body area not enlarged, homogeneous, without calcification, pancreatic tail obscured by intestinal gases difficult to assess, kidneys of normal and comparable size, without signs of stagnation and calcified deposits, unexpanded spleen. Magnetic resonance imaging of the abdomen was performed and revealed no significant changes, suggesting mild pancreatic injury.

In the hospital, the patient received paracetamol 1g *iv* and was discharged home (on her demand, despite indications for hospital treatment) with the recommendation of a strict diet (fluids only, 2–3 litres daily), gastro-resistant capsules of pancreatic enzymes (10,000 units orally, three times daily) and proton pump inhibitors (pantoprazole 40 mg orally, two times daily). Biochemical parameters were then assessed two days later (Table 2), showing a significant decrease in CRP (still above norm) and urine amylase (normal result). Moreover, slightly elevated total bilirubin was found. The analysis performed eight days later revealed correct CRP and urine amylase values and slightly increased total and direct bilirubin.

Because the patient is an actively working physician, it was concluded that the risk of serious complications after infection with SARS-CoV-2 outweighs the risk of potential mild pancreatic injury. As a result, the patient received a second dose of vaccine on her demand on 5 March 2021. No such severe reaction was observed at that time. The pain at the injection site was felt 2–3 h after vaccination. Twelve hours after receiving the vaccine, muscle and joint pains along with hyperesthesia appeared. Twenty-four hours after vaccination, lymphadenopathy under the left armpit was found, with lymph nodes enlarged to about 2–3 cm. The lymph nodes were painful for about four days.

## 3. Discussion

The described patient was a healthy woman, without any risk factors associated with the development of pancreatitis. She did not take any drugs or supplements and had no family history of pancreatitis. The pancreatic injury symptoms appeared shortly after the administration of the COVID-19 vaccine. Therefore, the temporal relationship may indicate the need for close monitoring of patients reporting abdominal pain after vaccination.

Vaccine-induced pancreatitis is an uncommon adverse reaction that was described for some viral vaccines such as Mumps–Measles–Rubella (MMR), Hepatitis A and B and Human papillomavirus [6,7,8]. Bizjak et al. identified in their paper important factors that suggest a possible link between vaccination and pancreatitis. These factors include [8]:the chronology of events associated with the onset of adverse reaction;positive rechallenge and exacerbation of symptoms after repeated exposure to the vaccine;similar case reports describing pancreatitis after using the same vaccine;a probable causal relationship of the vaccine to other kinds of autoimmune diseases;case reports describing pancreatitis observed after different vaccines;probable mechanism between the vaccine and acute pancreatitis.

According to Pfizer’s data, one case of pancreatitis and one obstructive pancreatitis adverse reaction was observed during Phase 2/3 clinical trial of COVID-19 mRNA vaccine [9]. The trial included about 38,000 participants, indicating that such a time link between vaccination and pancreatitis is a very rare adverse reaction. Similar information can be obtained from UK databases—a collection of spontaneous reports received between 9 December 2020 and 21 March 2021 for the mRNA Pfizer/BioNTech vaccine contains 40,883 reports of adverse reactions with one case of obstructive pancreatitis, three cases of pancreatitis, one case of acute pancreatitis and one case of necrotising pancreatitis [10].

The mechanism responsible for vaccine-induced pancreatitis remains unclear. Currently, the most probable hypothesis was proposed in the molecular mimicry theory, stating that amino acid sequences similarities between viral and self-antigens can result in autoimmune reaction [11]. Induction of such an autoimmune response can result in the production of cytotoxic antibodies with affinity for pancreatic acinar cells [8]. Indeed, recently published data revealed that antibodies against SARS-CoV-2 spike protein and nucleoprotein presented cross-reactivity against many human tissue antigens [12,13]. Other hypotheses that have been discussed include polyclonal activation of lymphocytes, “bystander activation” of self-reactive lymphocytes, somatic mutations of immunoglobulin variable genes, vaccine-induced vasculitis and triggering the release of histamine and leukotrienes [6].

Furthermore, pancreatitis can result from viral infection [14]. The scientific literature provides information on incidents of infectious pancreatitis caused by mumps (the first case of pancreatitis caused by virus ever described), hepatitis viruses (with hepatitis B virus being the most commonly associated with acute pancreatitis), human immunodeficiency virus (incidence of acute pancreatitis 40% in HIV-positive patients compared to 2% in the general population), coxsackie virus type B, herpes simplex virus, cytomegalus virus, varicella-zoster virus, and some other viruses [14,15]. Coronaviruses (CoVs) are a large family of single-stranded RNA viruses, infecting humans and animals and causing various organ symptoms [16,17]. It is worth noting that pancreatic injury was observed in 17% of COVID-19 cases [18]. Such injury might be caused by the direct cytopathic effect of SARS-CoV-2, as ACE2 receptors (used by the virus to infect the cell) were highly expressed in pancreas cells. However, another explanation underlines the role of systemic inflammatory and immune response in the development of pancreatitis during the COVID-19 course [18,19]. It is possible that a similar mechanism may be responsible for the adverse reaction observed in the described patient. The ongoing multi-centre observational study COVIDPAN, focused on acute pancreatitis during the COVID-19 course, together with retrospective analysis of vaccine adverse effects, may in the future provide data supporting this hypothesis [20].

## 4. Conclusions

To our knowledge, the presented case is the first report of acute pancreatitis, observed shortly after the COVID-19 vaccine, described in the scientific literature. It is consistent with the data from clinical trials, where only two cases of pancreatitis were reported in the group of almost 38,000 participants. The report shows that severe abdominal pain after vaccination should not be disregarded, as it can be a symptom of acute pancreatitis, and patients reporting this symptom should be closely monitored. Moreover, the analysis of the presented case can conclude that the presence of such adverse effect is not necessarily an indication for omitting the second dose of the vaccine—despite acute pancreatitis after the first vaccination, the patient received the second dose without triggering a severe response. However, such a decision should be taken carefully after a deep analysis of the patient’s observed adverse event and clinical condition.

## Figures and Tables

**Figure 1 vaccines-09-00576-f001:**
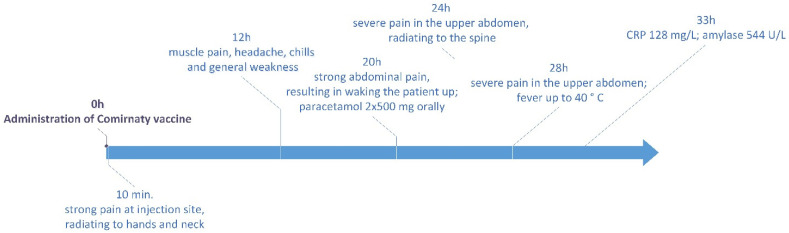
Adverse effects observed after administration of COVID-19 vaccine.

**Table 1 vaccines-09-00576-t001:** Blood morphology of the patient, assessed on 9 January 2021.

Parameter [Unit] (Reference Values)	Result
WBC [$10^{9}$/L] (4.00–10.00)	6.72
%NEUT [%] (45.0–70.0)	75.6 *
%LYMPH [%] (20.0–45.0)	16.1 *
%MONO [%] (0.0–9.0)	4.9
%EOS [%] (0.0–5.0)	1.3
%BASO [%] (0.0–1.5)	0.4
%LUC [%] (0.0–4.0)	1.7
#NEUT [$10^{9}$/L] (1.80–7.00)	5.08
#LYMPH [$10^{9}$/L] (0.80–4.50)	1.09
#MONO [$10^{9}$/L] (0.00–0.90)	0.33
#EOS [$10^{9}$/L] (0.00–0.50)	0.09
#BASO [$10^{9}$/L] (0.00–0.15)	0.03
#LUC [$10^{9}$/L] (0.00–0.40)	0.12
RBC [$10^{12}$/L] (4.00–5.00)	4.29
HGB [mmol/L] (7.45–10.00)	8.50
HCT [L/L] (0.36–0.47)	0.39
MCV [fL] (82–97)	90
MCH [fmol] (1.64–2.08)	1.98
MCHC [mmol/L] (20.00–22.00)	22.01 *
RDW [%] (11.00–15.00)	12.60
PLT [$10^{9}$/L] (130.0–390.0]	161
MPV [fL] (7.00–11.00]	9.40
PCT [L/L] (0.002–0.004)	0.002
PDW [%] (40.0–60.0)	67.5 *

* indicates results not in the reference range.

**Table 2 vaccines-09-00576-t002:** Results of biochemical analyses.

Parameter [Unit] (Reference Values)	9 January 2021	11 January 2021	19 January 2021
CRP [mg/L] (<5.0)	128.0 *	38.0 *	2.2
Urine amylase [U/L] (30–200)	544 *	208	240
Serum amylase [U/L] (25–115)	51	58	64
ALAT [U/L] (<45)	<7	8	8
ASPAT [U/L] (<35)	15	14	13
Serum creatinine [mg/dL] (0.50–0.90)	0.68	-	0.83
Potassium [mmol/L] (3.5–5.1)	3.8	-	4.85
Sodium [mmol/L] (135.0–145.0)	139.0	-	141.0
Fasting glucose [mg/dL] (70–99)	-	84	89
Total bilirubin [mg/dL] (<1.20)	-	2.21 *	1.24 *
Direct bilirubin [mg/dL] (<0.3)	-	-	0.40 *
Alkaline phosphatase [U/L] (35–105)	-	61	67
GGT [U/L] (5–36)	-	12	14
Total cholesterol [mg/dL] (<190)	-	174	-
HDL [mg/dL] (>45.0)	-	78	-
%HDL [%] (>20.0)	-	45	-
Triglycerides [mg/dL] (65–150)	-	45	-
Iron [µg/dL] (37–145)	-	80	-
UIBC [µg/dL] (112–346)	-	224	-
TIBC (calculated) [µg/dL] (250–400)	-	304	-
TSAT (calculated) [%] (20–55)	-	26	-
CEA [ng/mL] (<5.0/<6.5 smoking/non-smoking)	-	0.88	-
CA [U/mL] (<39)	-	4.4	-

* indicates results not in reference range.

## Data Availability

The data presented in this study are available on request from the corresponding author.

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
