# Peer review of "Pancreatic Injury after COVID-19 Vaccine—A Case Report"

_vaccines, 2021, doi:10.3390/vaccines9060576_

Round 1

Reviewer 1 Report

I was invited to revise the paper entitled "Pancreatic injury after COVID-19 vaccine – a case report". It was a case report reporting the occurrence of mild pacreatitis in a young healthy woman after the injection of Comirnaty vaccine. The paper is interesting and report a particular reacton after vaccination.

The paper appear complete, deeply reporting the case.

I have some observation:

  • Fig. 1 has low quality;
  • In discussion section authors should highlight that actually there is only a time link between vaccination and pancretitis. This is the principal limitation;
  • Authors should discuss the link between pancreatitis and HPV presented in the paper Bizjak, M., Bruck, O., Praprotnik, S. et al. Pancreatitis after human papillomavirus vaccination: a matter of molecular mimicry. Immunol Res 65, 164–167 (2017). https://doi.org/10.1007/s12026-016-8823-9;
  • Conclusions cannot report "indicating vaccine induced pancreatitis". There is only a time link and not a causative effect. Literature try to understand the link but actually we cannot conclude that vaccination cause pancretitis with certainity.

Author Response

Dear Reviewer,

thank you for your valuable comments that we will take into consideration to improve the quality of our work.

  1. Figure 1 was redrawn using bigger, more readable font and exported using print quality (600 dps).
  2. Thank you for this important remark. Indeed, the case presents only a time link between vaccination and pancreatitis, as there is no data explaining the mechanism. We did suggested correction in the discussion section.
  3. The paper of Bizjak et al. presents important factors that suggest a possible link between vaccination and pancreatitis. We added a paragraph discussing this issue.
  4. Thank you for your valuable remark. Indeed, conclusions cannot report "indicating vaccine induced pancreatitis", as there is only a time link without any mechanism showing the causative effect. The literature tries to understand this relationship, but we cannot really say that vaccination certainly causes pancreatitis. Further retrospective observations are needed while patients reporting such adverse effect should be closely monitored. We changed the conclusion section according to your comment.

Reviewer 2 Report

The Report 'Pancreatic injury after COVID-19 vaccine – a case report' is very interesting and provides insights into the ongoing vaccine drive all the countries have embarked upon. 

Even though the report clearly states pancreatic injury after COVID Vaccine, It is very important to consider abdominal pain after vaccination as an important symptom based on the report. However, the authors need to provide data that these symptoms are clearly related to the vaccine.

  1. The authors show that after 24 hrs of vaccine administration, there is a pancreatic injury. Can the authors provide a negative RT-PCR or Lungs CT Scan report prior to the vaccine administration. 
  2. It was good that they looked into CRP, did they perform biochemical assays to look into D-Dimer, Ferritin levels, to understand if the person had an asymptomatic COVID infection. 
  3. Since the patient was hospitalised, did they perform assays for analysing cardiac markers like Troponin, CK-MB.
  4. Did they perform assays for investigating IL-6 levels, to know if there is a inflammatory response or Cytokine Release Syndrome effects.
  5. It would be really interesting to understand the molecular mechanism of SARS-COV-2 activation of ACE2 receptors and their downstream pathways.
  6. The authors claim that the second dose did not have an effect on the subject. That is interesting, Can the authors give an explanation for it.

Minor

  1. A more impactful sentence is required. In Abstract give hope for controlling the pandemic.
  2. Comiranty mRNA (European COVID-19 vaccine synonymous to BioNtech Pfizer Vaccine).
  3. There a minor grammatical corrections in the report.

Author Response

Dear Reviewer,

thank you for your valuable comments that we will take into consideration to improve the quality of our work.

  1. Thank you for the important remark. The patient had RT-PCR COVID-19 test carried out with a negative result, excluding potentially ongoing COVID-19. Relevant information was added to the text. Lungs CT Scan was not carried out because it is not a routine emergency or hospital examination to be done in such a clinical situation. The patient reported to the hospital emergency during the weekend, where she received help and was referred to a gastroenterologist.
  2. Thank you for a valuable remark. D-Dimer and Ferritin assays were not carried out; as the result of RT PCR test was negative, asymptomatic COVID-19 was unlike.
  3. Cardiac markers were not assessed as there were no symptoms suggesting problems with the cardiovascular system.
  4. IL-6 levels were not measured because it is not a routine emergency or hospital examination to be done in such a clinical situation. However, it is a very important comment, as this information would provide interesting information on the cytokine inflammatory response. CRP level was assessed with significantly increased value.
  5. We agree completely. Knowledge of the mechanism of SARS-COV-2 activation of ACE2 could provide important information concerning potential means to counteract this process. The crucial role is played by viral S-protein, which has ACE2-binding domains. It was found that due to amino acid sequence changes, SARS-CoV-2 spike glycoprotein has 10 to 20-fold higher affinity to ACE2 compared to S protein of previous SARS-CoV (Medina-Enrizuez, 2020). A recent study by Anand et al. revealed that SARS-CoV-2 had a unique S1/S2 cleavage site that mimics a FURIN-cleavable peptide present in epithelial sodium channel α-subunit that may further enhance viral internalization capabilities.
  6. Thank you for your valuable remark. The decision to administer a second dose was a difficult one. We were not able to predict the body's immune response in terms of both humoral and cellular responses, so we took into account the risk of mild pancreatitis, but we recognized that the risk of developing COVID-19 and its potential complications in an actively working physician might be much more dangerous. Given the limited data from only one case, we cannot explain why a similar pancreatic injury reaction did not occur after the 2nd dose of the vaccine. Further research is needed in this area.
  7. Minor comments
    1. Small corrections were made in the abstract.
    2. We decided to change Comrinaty to a more informative "COVID-19 mRNA vaccine".
    3. Minor corrections were made in the text.

Reviewer 3 Report

Dear Authors,

I have perused with great interest the case report which you have submitted, concerning the alleged association between acute pancreatitis and the Comirnaty mRNA COVID-19 vaccine. You have accurately elaborated on the known linkages between acute pancreatitis and viral infections, but you have for some reason left out HIV-related AP, which has been reported to account for as much as 40% of HIV positive patients, as opposed to roughly 2% in the general population reference Rawla P, Bandaru SS, Vellipuram AR. Review of Infectious Etiology of Acute Pancreatitis. Gastroenterology Res. 2017;10(3):153-158. doi:10.14740/gr858w;  . That needs to be added in your discussion, when you bring up the viral etiology of AP, along with the other viral conditions which you have rightly mentioned. By the same token, I feel cytomegalovirus (CMV) and herpes simplex virus (HSV) should also be mentioned in that regard.
The hypothetical association between vaccines, and the Pfizer COVID-19 vaccine in particular, has not been expounded upon yet, and your list of hypothetical linkages is adequately presented and well fleshed out.  You may want to specify the possible correlation between AP and the production of cytotoxic autoantibodies affecting pancreatic acinar cells.
Overall, as a case-report, your work is a valuable and well-written contribution, the tables are well-assembled albeit a little too dense, and I believe it is worthy of publication.

Sincerely,

Author Response

Dear Reviewer,

thank you for your valuable comments that we will take into consideration to improve the quality of our work. As you suggested, we added information concerning the association of pancreatitis and HIV, CMV and HSV.

Indeed, the association between the Pfizer COVID-19 vaccine and pancreatitis is hypothetical. We believe that the presented clinical situation is a valuable observation that should be taken into account when carrying out undoubtedly necessary vaccinations in the general population.  Moreover, reporting such clinical situations, even if identified as a side effect, should be widely conducted.

A correlation between AP and the production of cytotoxic autoantibodies affecting acinar cells cannot be excluded. We added some information concerning this issue to the discussion.